# Analyzing and Modeling the Municipal Sewage Sludge Drying Process Using Python

**Erik Mihelič** [1,*], **Dušan Klinar** [1], **Klavdija Rižnar** [1] and **Primož Oprčkal** [2]

[1]   Scientific Research Centre, 2250 Ptuj, Slovenia; dusan.klinar@bistra.si (D.K.); klavdija.riznar@bistra.si (K.R.)
[2]   Slovenian National Building and Civil Engineering Institute (ZAG), 1000 Ljubljana, Slovenia; primoz.oprckal@zag.si
[*]   Correspondence: erik.mihelic@bistra.si

**Abstract:** The programming language Python offers the opportunity to analyze and model the municipal sewage sludge (MSS) drying process in an illustrative chemical engineering practice. The drying process is performed on a flat plate while maintaining a uniform, parallel drying air speed. The Python program helps to analyze the digitalized weight measurements from each sample. The program enables the sorting of input data, determination of the drying critical point, and evaluation of the first and second drying periods. Moreover, the model calculates the fundamental drying parameter and forms a drying master curve to support the transfer to different drying conditions. The basic parameters calculated are mass transfer coefficient, heat transfer coefficient, and diffusion coefficient. The results are consistent with published data for those coefficients over the drying temperature range of 19.4–52.4 °C and relative humidity range of 8.2–33.6%. The findings of this study demonstrate the potential of Python as a powerful tool for analyzing experimental data and modeling chemical processes, which can lead to improved process design, optimization, and control.

**Keywords:** data series processing; Python process modeling; sewage sludge drying; drying master curve





## 1. Introduction

Municipal sewage sludge (MSS) is generated as a by-product of wastewater treatment from wastewater treatment plants (WWTP). It presents a significant challenge to urban environments due to its high water content (around 80% water fraction), large volume, and the presence of various contaminants [1]. Proper treatment and preparation of the MSS to be incorporated into the Circular Economy (CE) are crucial for maintaining public health, environmental sustainability, and efficient resource management [2]. Various approaches have been employed to effectively manage MSS, focusing on agricultural soil amendments (either directly or following composting), land remediation, incineration, and heat generation, producing biochar and construction composites, extracting phosphorus, and producing fertilizers. Drying usually represents one of the initial steps of these processes. Appropriate design and selection of drying technology for MSS are crucial for sustainable and economical utilization. A significant aspect of this project revolves around producing a dried MSS product with around 30% moisture content suitable for agricultural applications. While debates persist, particularly regarding organic pollutants, numerous countries prefer utilizing MSS in agriculture due to its positive impact on soil quality, attributed to the presence of organic matter and the nitrogen (2.8–3.8%) and phosphorus (2–4%) content in MSS [3,4]. Python offers a user-friendly syntax and robust libraries for data manipulation, statistical analysis, machine learning, and visualization, enabling researchers to handle complex datasets and derive meaningful insights efficiently. Furthermore, its open-source nature facilitates collaboration and reproducibility in scientific endeavors, supporting and encouraging digitalization. In the context of chemical engineering, Python has emerged as a powerful tool for modeling, simulation, data analysis, optimization, control, and more.

Although limited literature exists on the direct use of Python for drying analysis, Python has been extensively utilized in environmental engineering and modeling, including wastewater treatment processes [5,6] and others like pyrolysis and extraction. The paper provides an outstanding example of interdisciplinary research, combining programming and chemical technology. A digitalized laboratory and other measuring equipment give large series of processed data which are in most cases relatively scattered but contain vital process or material specific information. As chemical engineers are mostly familiar with the process or chemistry mechanisms behind a material and/or process, specific parameters can be extracted from these large series of data. Some specific examples of Python applications in chemical engineering are summarized below.

Modeling and simulation: Python's powerful libraries, such as NumPy and SciPy, enable engineers to build mathematical models and simulate chemical processes, including drying, distillation, reaction kinetics, and heat transfer. This helps to optimize process design and troubleshoot potential problems.

Data analysis and visualization: Python's data analysis libraries, such as Pandas and Matplotlib, facilitate the analysis of experimental data, quality control, and visualization of process parameters. This helps engineers to gain insights into process performance and identify areas for improvement.

Optimization: Python's optimization libraries, such as SciPy and PuLP, can be used to optimize chemical processes, including parameter tuning, minimizing energy consumption, and maximizing product yield. This can lead to significant cost savings and environmental benefits.

Machine learning and predictive analytics: Python's machine learning libraries, such as scikit-learn and TensorFlow, enable predictive maintenance, anomaly detection, and real-time monitoring of chemical processes. This helps to improve process efficiency and reduce downtime.

Control systems: Python can be used to develop control algorithms and implement process control systems. Libraries like control and scipy.signal are useful for designing controllers to maintain process variables within desired ranges. This ensures the stable and efficient operation of chemical processes.

MSS drying techniques can be broadly classified according to heat transfer principles (convective, conductive, radiation, or mixed), contacting (disc, thin film, and drum), and transport methods (fluid bed, pneumatic, belt, screw, vacuum, and others) [7–9]. The resulting water vapors are usually discharged by drying gas (air or flue gases). MSS drying targets different water contents (<85% DM) declared as partial drying and complete drying (>85% DM) and this determines the type of dryer selected. Capacity plays an important role in the final decision; in Germany [9], mainly drum, belt, and solar dryers are used according to the capacity needed. Drying parameters reported include drying temperature (drum dryer, 85–115 °C; belt, 60–70 °C; and solar, 10–40 °C), final moisture content (<10% and 30–50% with solar drying), initial moisture content (75–80%), and various residence times, which are crucial in determining the energy efficiency and overall performance of the drying process [8]. A key decision in equipment selection is the source of heat, as it should be low cost or even costless (e.g., solar heating). Expensive heat sources usually generate a problem of high OPEX and can even overcome allowable costs. Researchers have also explored the potential of combining different drying methods to enhance efficiency and reduce energy consumption [9]. Additionally, studies have examined the environmental impact (e.g., odor emissions) of different drying methods and their implications for sustainability [10].

Our paper adds value to the expanding field of drying processes in the academic community, with a specific focus on Python programming. Currently, the global literature predominantly centers on Python's involvement in drying, with significant recognition given to the 'pydring' package [11]. This package utilizes multiphysical and dynamic modeling of drying phenomena, incorporating finite volume numerical methods to solve drying equations and analyze drying kinetics for a range of solid products. It is extensively utilized in food processing, biomass refining, and animal feed production. However,

our study highlights a crucial gap in the literature. While 'pydring' and related research investigate fundamental drying parameters, they fail to consider the specific demands of MSS drying. Our study introduces an innovative program designed for MSS drying, enabling the collection of a wider range of data, encompassing mass flux, thermal flux, and critical points. As a result, we fill an existing gap in the field and contribute new insights that advance the understanding of MSS drying. Notably, our program's distinctive attribute is its ability to carry out regression analysis on the initial and subsequent drying periods, establish regression functions, and identify crucial drying points. During our scientific literature review, we came across various drying scenarios in which Python programming was used. These included the drying of organically grown apples [12], the process of drying yellow passion fruit seeds [13], pharmaceutical granule drying with a focus on kinetics [14], and predicting coal moisture content during convective drying [15]. However, no previous research has connected Python to the specific field of MSS drying, emphasizing the original and multidisciplinary nature of our study, which combines programming with the chemical–technological complexities of sludge drying.

This research paper presents a comprehensive study on the design and analysis of the MSS drying process, utilizing the Python programming language to process experimental data and calculate essential drying parameters [16].

## 2. Materials and Methods

In our research, the drying experiment was conducted using a digitalized weighing scale, enabling continuous monitoring and recording of the MSS sample's weight at fixed and short time intervals [17]. The novelty of this study lies in the first attempt to develop a program using the Python programming language with the aim of successfully analyzing measurements of chemical–technological processes such as drying processes of MSS in our case. This demonstrates that Python is a suitable tool for analyzing such data. This approach allowed accurate tracking of the drying process, providing valuable insights into the behavior of MSS under varying conditions. This study was carried out using five samples of MSS. Each sample underwent an analysis process that was repeated three times for thorough validation, and no discrepancies were observed. The experimental conditions were held constant at an airspeed of 1.15 m s$^{-1}$ and a range of temperatures, including 19.4, 22.0, 29.0, 44.0, and 52.4 °C [17–19]. In the following segment, we will present a more detailed description and operation of the program, the theoretical background of the model, and the results we obtained from the drying analysis of the five MSS samples.

The experiment part was carried out in the laboratory of ZRS Bistra Ptuj, utilizing state-of-the-art digital equipment to ensure accurate and reliable results. Municipal sewage sludge (MSS) samples were obtained from the Ptuj municipality sewage treatment plant, ensuring that the samples were fresh and representative of the typical composition of the sludge in the region. Drying was performed on a digitized G&G electronic scale (model: JJ200B), allowing precise measurement and data collection throughout the process. Weighing data were recorded on a computer via USB at a fixed time interval of 20 s, ensuring a comprehensive and consistent record of the mass changes during the drying process. The MSS was dried using a LINEA portable air heater dryer (model: LP1-0521), which provided controlled and consistent drying conditions for each experiment. The airflow was measured with an RS PRO turbine anemometer, which allowed accurate measurement and documentation of the airflow conditions throughout the drying process.

### 2.1. Experimental Procedures for Drying Characterization of the MSS Samples

The experimental procedure involved the following steps:

1. An MSS sample was carefully distributed on a flat tray (surface area: 56.7 cm$^2$ and precise height, $L = 1.5$ mm) using a leveling knife. Prior to the application of MSS, a circular metal mesh (1 mm square opening) was inserted into the tray to ensure even distribution of the sludge during the drying process due to extensive shrinkage. This step was crucial to obtain accurate and reproducible results by minimizing variations

in the surface of dry MSS that could impact the drying process because of enormous shrinkage.

2. Weighing variations caused by air blowing from the dryer parallel to the sample surface were accounted for via blind measurement. The blind measurement ran for 5 min. Mass readings were recorded every second, and the results were used to correct weighing results for absolute error by subtracting from the data for further analysis. This step ensured that the results precisely reflected changes in mass due to the drying process rather than the influence of airflow on the weighing process.

3. Before placing the MSS sample on the balance, the air flow was measured using an anemometer. Measurements help to ensure consistent drying conditions across all experiments and accurately document the conditions under which the drying occurred. Drying was conducted for 2–6 h, depending on temperature and humidity, to ensure completion of the drying process and accurate representation of the drying behavior of MSS under the given conditions.

4. After the drying process, the final mass of MSS and other parameters, such as relative humidity and dry bulb temperature, were recorded. The airflow was measured again to ensure consistency in drying conditions throughout the experiment. Comprehensive documentation of the experimental conditions enabled thorough analysis and interpretation of the results.

5. The entire procedure was conducted at five different drying temperatures, ranging from 19.4 to 52.4 °C, with a constant air speed of 1.15 m s$^{-1}$. Throughout the drying process, relative humidity was measured, and its value was found to be highly dependent on temperature.

6. All weighing data were transferred to an Excel file, from which a Python program read the data for further analysis. This approach enabled efficient processing and analysis of the data, ensuring thorough examination and accurate interpretation of the results. Standardized data formats and widely used software tools also ensured transparency, reproducibility, and adherence to the principles outlined in the text above.

### 2.2. Python Program Explained

The Python-based program used in this study consists of five main phases. In Stage 1, the experimental data are sorted and averaged to generate a robust dataset for analysis. Stage 2 involves conducting regression analysis on the data from the initial drying period, resulting in a linear function that represents the first drying period. Stage 3 focuses on regression analysis of the data from the second drying period, yielding an exponential function and identifying the critical drying point [20]. Stage 4 of the process involves the Python program computing drying parameters to assess the efficiency and effectiveness of the drying process. These calculated parameters are crucial for understanding various aspects and dynamics of the drying procedure. Subsequently, in Stage 5, the program performs a dual function: enable graphical visualizations of the data, facilitating a clear depiction and comprehension of the results, while simultaneously exporting these results to a separate Excel file. This export function enables comprehensive data analysis and ensures that the results are readily available for further examination and utilization, thereby enhancing the reproducibility and accessibility of the research.

### 2.3. Use of External Libraries

It is important to note that the Python program leverages on external libraries to streamline data processing and analysis. These libraries include pandas for data manipulation and openpyxl for handling excel files, math for mathematical functions, matplotlib.pyplot for data visualization, numpy for numerical operations, matplotlib.mlab for additional plotting tools, sklearn.linear_model for regression analysis, and scipy.optimize for curve fitting [21].

### 2.4. Python Limitations and Comparison to Other Software

This subsection discusses the limitations of Python and compares it to other popular programming languages and software. Table 1 provides a summary of the key differences between Python and other software.

**Table 1.** Python Limitations and Comparison to Other Software.

| Software | Functions | Applications | Limitations |
|---|---|---|---|
| PYTHON | Python is a versatile programming language that excels in various functions, including moisture content calculation, drying rate model, drying time estimation, energy consumption calculation, regression analysis for both first and second drying periods, and seamless data visualization. | Python can be applied for convective drying modeling in various ways, including simulating drying kinetics, optimizing drying processes, estimating energy consumption, predicting drying times, visualizing data, using machine learning for predictions, performing sensitivity analysis, conducting comparative analyses, enabling real-time monitoring and control, and creating educational tools. | Limitations in using Python for convective drying modeling include potential challenges in handling large datasets, computational intensity for complex simulations, the need for specialized libraries or expertise, and limitations in real-time control applications. |
| COMSOL Multiphysics (Burlington, MA, USA) | This software serves as a simulation tool, offering insights into the heat transfer profile and fluid flow patterns within the dryer. | This software has the capability to predict the airflow from the inlet to the outlet and determine the precise location of ventilation holes. Additionally, it can be employed to anticipate the exact shape and dimensions of the dryer. | In comparison to CFD FLUENT, learning this software is relatively straightforward. |
| FORTRAN (Armonk, NY, USA) | It finds application in simulation and modeling to address and solve partial differential equations. | This software is valuable for conducting performance analyses of convective drying systems, resulting in potential cost savings through efficient material usage. It optimizes structural performance with in-depth analysis and eliminates the need for expensive and time-consuming trial-and-error processes. | The Fortran program is initially developed in a prototype software, often utilizing visual languages like Matlab and IDL (Interactive Data Language). Subsequently, the code is ported to FORTRAN for further development and implementation. |
| MATLAB (Natick, MA, USA) | MATLAB serves as a mathematical modeling software, enabling highly accurate and efficient nonlinear regression analysis within a short timeframe. | This software is highly beneficial for creating mathematical models to predict crop temperature, air temperature, and moisture evaporation. It is also a valuable tool for training and testing various models. | MATLAB mathematical modeling demands strong programming skills, and the development and testing of models can be a time-consuming process. |
| Sigma Plot (San Jose, CA, USA) | It is an analytical software. | Simulation validation involves assessing various statistical parameters for greenhouse drying performance, including factors such as moisture evaporation rate and greenhouse room temperature. | This software often involves repetitive analysis as a common characteristic. |
| TRNSYS (Madison, WI, USA) | TRNSYS stands as a versatile scientific simulation tool in the field of convective energy. Its software is instrumental in the development and description of crop drying behavior within various types of dryers. | A significant advantage lies in its ability to replace complex differential equations with straightforward numerical calculations. This software simplifies the calculation of moisture and heat transfer at the crop surface to the drying air, as well as the transport of moisture and heat within the crop's interior. | More accurate results are achieved when using shorter time steps and when the segments are closely aligned in every numerical method. |

### 2.5. Stage 1: Data Sorting

In our study, the accurate analysis of drying parameters is crucial, particularly when dealing with empirical data obtained from experimental measurements of the mass of

the MSS during drying. These measurements were recorded using a balance. Due to the convective drying process, the recorded values exhibited fluctuations over time. To address this variability and ensure a more regular representation of the data, it was necessary to sort and average the measurements. This averaging process aimed to create a more consistent curve showing how the mass varies with time and better capture the overall trend. The process involved selecting a certain number of measurement points ($N$), arranging them, and computing the mean value based on a given equation during the initial phase of data analysis.

$$y = \frac{y_1 + y_2 + \ldots + y_n}{N} \tag{1}$$

The software enables us to adjust the value of $N$ according to our preferences. Decreasing $N$ results in more scattered data, whereas increasing $N$ provides more structured data that align better with the trend line of the drying curve. Nevertheless, exercising caution is necessary to avoid $N$ becoming too large, as it could cause a significant deviation in the average data from the actual value. After careful analysis, we determined that an $N$ value ranging from 5 to 10 offers an ideal balance between data smoothing and preservation of the original measurements. Additionally, it guaranteed precise computation of the drying parameters based on initial data. Figure 1 presents a comparison of sorted and unsorted mass data over time and the first mass derivative with respect to time.

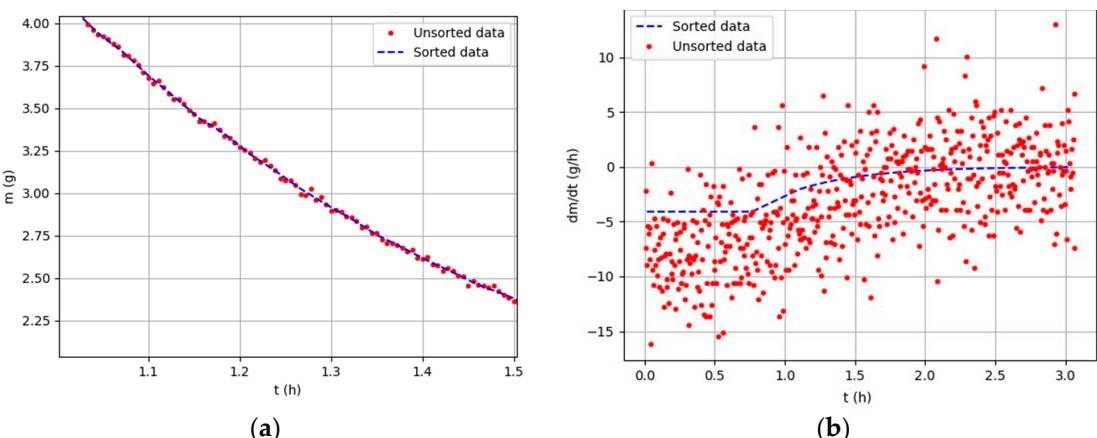

**Figure 1.** Sorted vs. unsorted data: (**a**) mass change over time; (**b**) first derivative over time.

### 2.6. Stage 2: Linear Regression of the First Drying Period

The first drying period, also known as the constant speed drying period, occurs at the beginning of the drying process when the drying speed is constant. During this period, there is sufficient moisture on the surface of the material and the evaporation rate is mainly determined by external conditions such as temperature, pressure, humidity, and air drying rate [22]. Since the time dependence of the mass reduction is linear, the 1st drying period can be given by Equation (2):

$$f_1(t) = k_1 t_1 + n \tag{2}$$

In Stage 2 of the program, the critical drying point is determined by analyzing the regression of the first drying period [18]. The program uses a specific algorithm that relies on a key criterion: $R^2$ must be greater than 0.99. By applying this criterion, the program can identify the final point of the first drying period, determining the coordinate of the critical moisture content of the sample. Once the $R^2$ value falls below 0.99, this point is identified as the critical drying point. The program also links this critical point with the start of the second drying phase and the beginning of Stage 3 [22].

### 2.7. Stage 3: Second Drying Period Analysis

Stage 3 is based on the exponential regression function derived in Section 2, specifically in the subsection titled "Derivation of the Second Drying Period's Regression Function," using experimental data from the second drying period.

$$f_2(t) = Ae^{-k_2 t_2} \tag{3}$$

The curve representing the second drying period follows an exponential function in the form of Equation (3). The exponential function captures the falling rate period of the drying process as the rate of moisture loss decreases over time. In the following sections, we will delve deeper into the derivation and details of this exponential function, exploring its implications and applications in the context of drying processes.

### 2.8. Derivation of the Second Drying Period's Regression Function

To derive the regression function of the second drying period, it is necessary to first define the basic drying parameters as follows:

$$X_t = \frac{m_t - Ls}{m_0} \tag{4}$$

where:

$X_t$—moisture content of the substance (g g$^{-1}$);
$m_t$—mass of the sample, which varies with time (g);
$Ls$—initial mass of the dry matter (g),

This equation calculates the moisture content of the substance at any given time depending on mass of the sample, which varies with time ($m_t$). The moisture content decreases as the drying process continues [22].

Free moisture is calculated as follows:

$$X_f = X_t - X^* \tag{5}$$

where:

$X_f$—free moisture content of the substance (g g$^{-1}$);
$X^*$—equilibrium moisture content at given constant drying conditions (g g$^{-1}$).

Free moisture represents the amount of moisture in the substance that exceeds its equilibrium moisture content ($X^*$). During the constant-rate drying period, the free moisture is available for evaporation, and the drying process is typically limited by the external factors mentioned earlier [22].

In our analysis, we assumed that the drying process during the second drying period can be characterized as pseudo-steady-state diffusion. Pseudo-steady-state diffusion refers to a scenario where the internal moisture diffusion within the material dominates the drying process, while external factors like surface evaporation and convective heat transfer have minimal influence. This assumption enables us to concentrate on the internal moisture transport mechanisms and simplifies the mathematical modeling of the drying process during the falling rate period. Pseudo-steady-state diffusion has been widely utilized in drying research to describe the behavior of different materials during the falling rate period [22–26]. The drying model and basic parameters are shown in Figure 2.

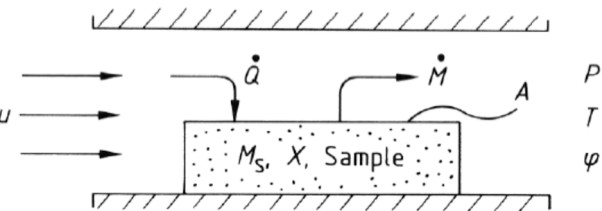

**Figure 2.** Basic parameters of the drying process.

To derive a function describing the free moisture content of a material during the second drying period, we start with a simple mass balance equation, which states that the difference in water mass flow is equal to the accumulation:

$$\Phi_{\text{in}} - \Phi_{\text{out}} = Accumulation \tag{6}$$

The mass flow of water can be expressed as follows:

$$\Phi = N_A M_w A \tag{7}$$

where:

$N_A$—mass flux of water (g s$^{-1}$ m$^{-2}$);
$M_w$—molar mass of water (18 g mol$^{-1}$);
$A$—surface area exposed to the drying air flow (m$^2$).

Assuming that during the drying process, the mass flow of water originates exclusively from the material, then therefore $\Phi_{\text{in}} = 0$, and the accumulation is equal to the change in mass over time. The water mass balance for the material system is as follows:

$$-N_A M_w A = \frac{dm_1}{dt} = \frac{d(V_1 M_w C)}{dt} = V_1 M_w \frac{dC}{dt} \tag{8}$$

Similarly, a mass balance of water can be derived for the air system, where it is assumed that all of the mass of water is transferred from the material to the air. $\Phi_{\text{out}}$ is simplified as follows:

$$\Phi_{\text{out}} = N_A M_w A = \frac{dLs}{dt} = \frac{d(V_0 M_w C_0)}{dt} = V_1 M_w \frac{dC_0}{dt} \tag{9}$$

Assuming the concentration of water in the air remains constant, then $dC_0/dt = 0$. The equations for the material and air systems are summed and rearranged, and the following is obtained:

$$-N_A A = V_1 \frac{dC}{dt} \tag{10}$$

Considering pseudo-steady-state diffusion and a thin film, the bulk flux can be approximated by the following equation [16]:

$$N_A = \frac{D_{AB}}{L}(C - C_0) \tag{11}$$

where:

$D_{AB}$—diffusion coefficient of water (m$^2$ s$^{-1}$);
$L$—film thickness (m).

Combining the above equations yields a first-order differential equation:

$$-\frac{D_{AB} A}{L V_1}(C - C_0) = \frac{dC}{dt} \tag{12}$$

Introducing a parameter k, where $k_2 = \frac{D_{AB} A}{L V_1}$, the solution to the differential equation at initial conditions $C_1(t_c) = C_1$ is as follows:

$$C(t) = (C_1 - C_0)e^{-k_2(t - t_c)} + C_0 \tag{13}$$

where:

$t_c$—the critical time where the second drying period starts (h);
$C_0$—initial concentration of water in MSS (mol L$^{-1}$);
$C_1$—concentration of water in the air (mol L$^{-1}$).

To determine how the moisture content of a substance, $X_t$, varies with time, we can express the water concentration as follows:

$$C = \frac{m_{H_2O}}{M_w V} \tag{14}$$

Combining Equations (14) and (4) gives the following equation:

$$X_t = \frac{M_w V C - Ls}{Ls} \tag{15}$$

To express $C$, the following equation can be obtained:

$$C = \frac{Ls(1 + X_t)}{M_w V} \tag{16}$$

Inserting the above equation into the previous one, we obtain the following:

$$\frac{Ls(1 + X_t(t))}{M_w V_1} = \left[ \frac{Ls(1 + X_t(t_c))}{M_w V_1} - \frac{Ls(1 + X_t(\infty))}{M_w V_1} \right] e^{-k_2(t - t_c)} + \frac{Ls(1 + X_t(\infty))}{M_w V_1} \tag{17}$$

where:

$X_t(t)$—moisture content of water changing over time (g g$^{-1}$);
$X_t(t_c)$—moisture content of water at point (g g$^{-1}$);
$X_t(\infty)$—moisture content of water at infinite time (g g$^{-1}$).
Rearranging Equation (17) gives the following:

$$X_t(t) = [X_t(t_c) - X_t(\infty)]e^{-k_2(t - t_c)} + X_t(\infty) \tag{18}$$

Subtracting the equilibrium humidity $X^*$ from both sides of Equation (18) gives the following:

$$X_t(t) - X^* = [X_t(t_c) - X_t(\infty)]e^{-k_2(t - t_c)} + X_t(\infty) - X^* \tag{19}$$

Considering the assumption that the humidity of a substance at infinite time $X_t(\infty)$ is equal to the equilibrium humidity $X^*$, the critical humidity is defined as follows:

$$X_c = X_t(t_c) - X^* \tag{20}$$

Taking into account the above equation and the assumption, we can write the final equation describing the variation in free moisture content of the material during the second drying period as a function of time [16,22–26]:

$$X_f(t) = X_c \, e^{-k_2(t - t_c)} \tag{21}$$

### 2.9. Stage 4: Calculating Drying Parameters

In Stage 4, the program calculates the drying parameters, which serve as key indicators for evaluating the efficiency and effectiveness of the drying process [22].

These parameters are as follows:

$$R_s = -\frac{Ls}{A} \frac{dX}{dt} \tag{22}$$

where:

$R_s$—the drying rate (g m$^{-2}$s$^{-1}$);
$A$—the area exposed to the drying air flow (cm$^2$).
Additionally, the program calculates the heat flux ($q$) according to the following equation:

$$q = hA\left(T_{dry} - T_{wet}\right) \tag{23}$$

where:

$q$—heat flow (W);

$h$—heat transfer coefficient (W m$^{-2}$K$^{-1}$);

$T_{\text{dry}}$—dry bulb temperature (°C);

$T_{\text{wet}}$—wet bulb temperature (°C).

Alternatively, $q$ is calculated as follows:

$$q = M_{\text{w}}\, N_{\text{A}}\, \Delta H_{\text{l,g}} A \tag{24}$$

where:

$\Delta H_{\text{l,g}}$—latent heat of vaporization (kJ kg$^{-1}$).

By rearranging Equations (23) and (24), we can express the heat transfer equation as follows:

$$h = \frac{q}{A\left(T_{dry} - T_{wet}\right)} \tag{25}$$

The mass flux ($N_{\text{A}}$) is calculated using the following equation:

$$N_{\text{A}} = k_{\text{y}} \frac{M_{\text{B}}}{M_{\text{A}}} \left(X_{\text{wet}} - X_{\text{dry}}\right) \tag{26}$$

where:

$M_{\text{B}}$—the molar mass of air (29 g mol$^{-1}$);

$k_{\text{y}}$—the mass transfer coefficient (mol m$^{-2}$s$^{-1}$);

$X_{\text{wet}}$—wet bulb humidity (g g$^{-1}$);

$X_{\text{dry}}$—dry bulb humidity (g g$^{-1}$).

From Equation (26), we can derive the mass transfer coefficient:

$$k_{\text{y}} = \frac{M_{\text{A}}}{M_{\text{B}}} \frac{N_{\text{A}}}{\left(X_{\text{wet}} - X_{\text{dry}}\right)} \tag{27}$$

The literature frequently presents the mass transfer coefficient in units of m s$^{-1}$. To compute the coefficient in these units, the following expression should be used:

$$N_{\text{A}} = k'_{\text{y}} \left(C_{\text{wet}} - C_{\text{dry}}\right) \tag{28}$$

where:

$k'_{\text{y}}$—mass transfer coefficient (m s$^{-1}$);

$C_{\text{wet}}$—concentration of water at wet bulb temperature (mol L$^{-1}$);

$C_{\text{dry}}$—concentration of water at dry bulb temperature (mol L$^{-1}$).

An example of the derivation of the water vapor concentration equation is given by the example of the water vapor concentration at wet bulb temperature. The derivation of the dry bulb water vapor concentration is similar.

It is necessary to use the gas equation:

$$C_{\text{wet}} = \frac{P_{\text{wet}}}{R T_{\text{wet}}} \tag{29}$$

where:

$R$—gas constant (8.314 J mol$^{-1}$K$^{-1}$);

$P_{\text{wet}}$—partial vapor pressure at wet bulb temperature (kPa).

The partial pressure of water vapor can be expressed in terms of the following equation [22]:

$$X_{\text{wet}} = \frac{M_{\text{A}}}{M_{\text{B}}} \frac{P_{\text{wet}}}{P - P_{\text{wet}}} \tag{30}$$

where:

*P*—atmospheric pressure (101.32 kPa).

Combining Equations (29) and (30) and highlighting $C_\text{wet}$ give the following equation:

$$C_\text{wet} = \left[ \frac{P \frac{M_\text{B}}{M_\text{A}} X_\text{wet}}{1 + \frac{M_\text{B}}{M_\text{A}} X_\text{wet}} \right] \frac{1}{RT_\text{wet}} \tag{31}$$

To calculate the mass transfer coefficient in units of m s$^{-1}$, we combine Equations (28) and (31) and calculate $C_\text{dry}$ via analogy with $C_\text{wet}$:

$$k'_\text{y} = \frac{N_\text{A}}{\left[ \frac{P \frac{M_\text{B}}{M_\text{A}} X_\text{wet}}{1 + \frac{M_\text{B}}{M_\text{A}} X_\text{wet}} \right] \frac{1}{RT_\text{wet}} - \left[ \frac{P \frac{M_\text{B}}{M_\text{A}} X_\text{dry}}{1 + \frac{M_\text{B}}{M_\text{A}} X_\text{dry}} \right] \frac{1}{RT_\text{dry}}} \tag{32}$$

Once these parameters are determined, the program systematically logs and writes them in an Excel file for convenient data management and subsequent analysis [22–27]

### 2.10. Stage 5: Graph Plotting

In stage 5, the program enhances data analysis by generating various diagrams. These include graphs that compare experimental and sorted data, plots of regression functions for the first and second drying periods against sorted data, and an overall drying curve plot that compares regression functions with experimental data. Additional diagrams depict the first derivative of humidity against time and the drying rate versus moisture content. Furthermore, a normalized drying rate curve is plotted using coordinates calculated using the following formulas [28]:

$$\nu = \frac{R_\text{s}}{R_\text{max}} \tag{33}$$

$$\xi = \frac{X_\text{t} - X^*}{X_\text{c} - X^*} \tag{34}$$

where:

$\nu$—relative drying rate (dimensionless);

$\xi$—characteristic moisture content (dimensionless);

$R_\text{max}$—maximum drying rate (g m$^{-2}$s$^{-1}$).

These graphical representations enhance the comprehension of the drying process and can be applied at various temperatures and air moisture conditions [22].

## 3. Results and Discussion

Table 2 displays the fundamental measurements that form the basis of the software's subsequent calculations. These measurements include the initial mass of the MSS, determined before the drying process. Additionally, we incorporated the final mass of the MSS, determined at the end of the drying process, and key parameters such as dry bulb temperature, relative humidity, surface area, and airspeed. The table reveals an inverse correlation between relative humidity and temperature (the temperature increases and relative humidity decreases); as the temperature of the same humid air is heated to higher temperature, the relative humidity becomes lower.

**Table 2.** Measured data for sample 1–5.

| Sample | $m_\text{MSS,start}$ (g) | $m_\text{MSS,end}$ (g) | $A$ [cm$^2$] | Drying Temperature [°C] | $T_\text{wet}$ [°C] | Ψ [%] | Airspeed (m s$^{-1}$) |
|---|---|---|---|---|---|---|---|
| 1 | 11.24 | 2.09 | 56.74 | 19.4 | 10.5 | 33.6 | 1.15 |
| 2 | 11.52 | 1.95 | 56.74 | 22.0 | 10.9 | 22.4 | 1.15 |
| 3 | 15.81 | 3.02 | 56.74 | 29.0 | 13.7 | 14.8 | 1.15 |
| 4 | 12.29 | 2.20 | 56.74 | 44.0 | 20.5 | 10.0 | 1.15 |
| 5 | 11.34 | 1.99 | 56.74 | 52.4 | 23.8 | 8.2 | 1.15 |

### 3.1. Stage 1 Results

The outcomes of stage 1 are demonstrated in Figure 3. For the sake of clarity, we have decided to show only the results of sample 1 and sample 5. The results of samples 2, 3, and 4 are in the Supplementary Materials. The graphs illustrate the changes in sample mass throughout the drying process. Each graph also shows a comparison between the sorted and unsorted data. This significant difference may not be visually apparent, but it is essential for subsequent calculations. The curves on the diagrams display a distinctive shape, indicating two distinct drying phases. The first phase is characterized by a decrease in sample mass proportional to time, while the second phase is identified by a reduction in the rate of drying. The length of the initial drying phase differs among samples due to the varying drying conditions applied to each one. Sample 1 experienced the longest first drying period phase due to being dried at 19.4 °C, while sample 5 had the shortest first drying period with a temperature of 52.4 °C (graphs for samples 2, 3, and 4 are in the Supplementary Materials, Figure S1).

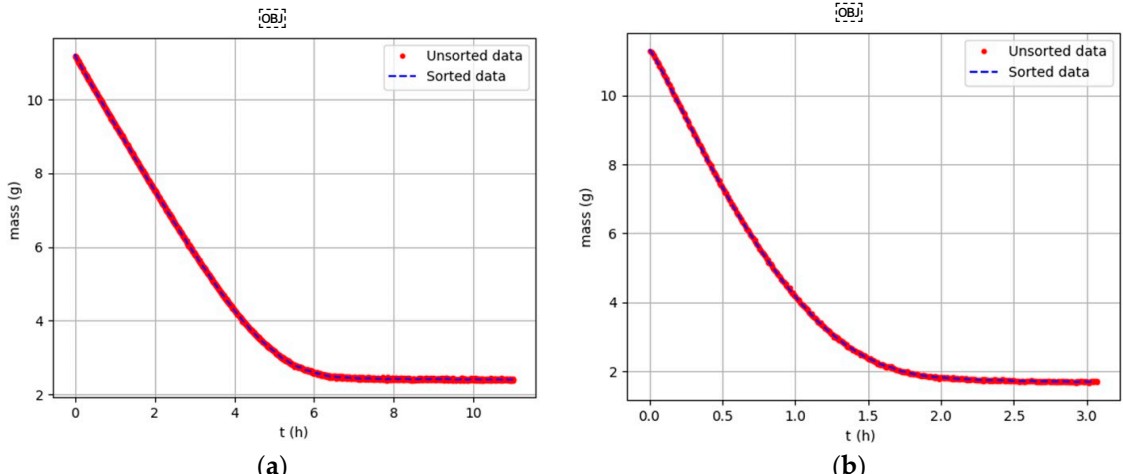

**Figure 3.** Mass of MSS over time: (**a**) sample 1, $T_{\text{dry}}$ = 19.4 °C, $\Psi$ = 33.6%; (**b**) sample 5, $T_{\text{dry}}$ = 52.4 °C, $\Psi$ = 8.2%.

### 3.2. Stage 2 Results

The outcomes of the second stage are demonstrated in Figure 4. The graphs exhibit the linear regression of each sample from the initial drying period. The plots are remarkably similar to one another, with an $R^2$ value of 0.996 for all of them. The equality of $R^2$ is because the program needed to establish an upper limit on $R^2$ as a condition for calculating the linear regression. The second stage ends, and the program enters the third stage when the $R^2$ value of the linear regression falls below 0.996. The diagrams presented in Figure 4 indicate that the slope of the lines between sample 1 and sample 5 becomes steeper as the rate of drying in the first period increases. This leads to shorter drying times, since sample 5 has fewer experimental points on its graph as compared to the preceding samples (graphs for samples 2, 3, and 4 are in the Supplementary Materials, Figure S2).

### 3.3. Stage 3 Results

The results of stage 3 are presented in Figures 5 and 6. Figure 5 displays the exponential regression of each sample during the second drying period. The variability in the results is primarily due to differences in the $R^2$ values. In contrast to stage 2, where $R^2$ was a fixed value, stage 3 involves a calculated $R^2$ value that depends on the fit of the experimental data and trend line. The diagrams reveal that some moisture content values drop below zero towards the end of the second drying period, a theoretically impossible occurrence. The fluctuations in mass measurements recorded by the balance are due to the fan air blowing parallel to the sample.

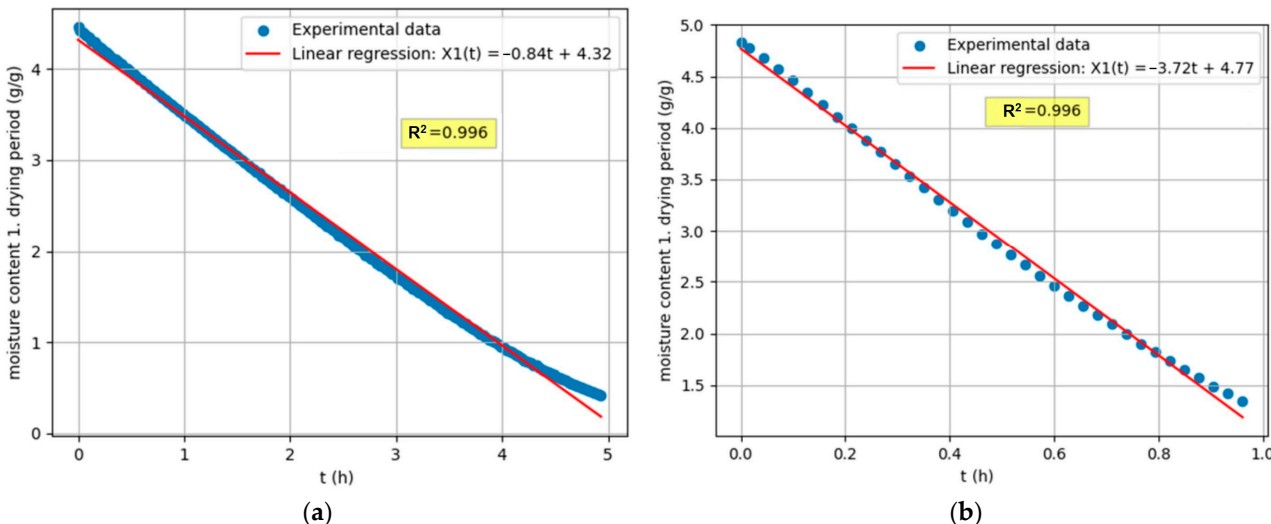

**Figure 4.** Regression functions of the first drying period: (**a**) sample 1, $T_{dry}$ = 19.4 °C, $\Psi$ = 33.6%; (**b**) sample 5, $T_{dry}$ = 52.4 °C, $\Psi$ = 8.2%.

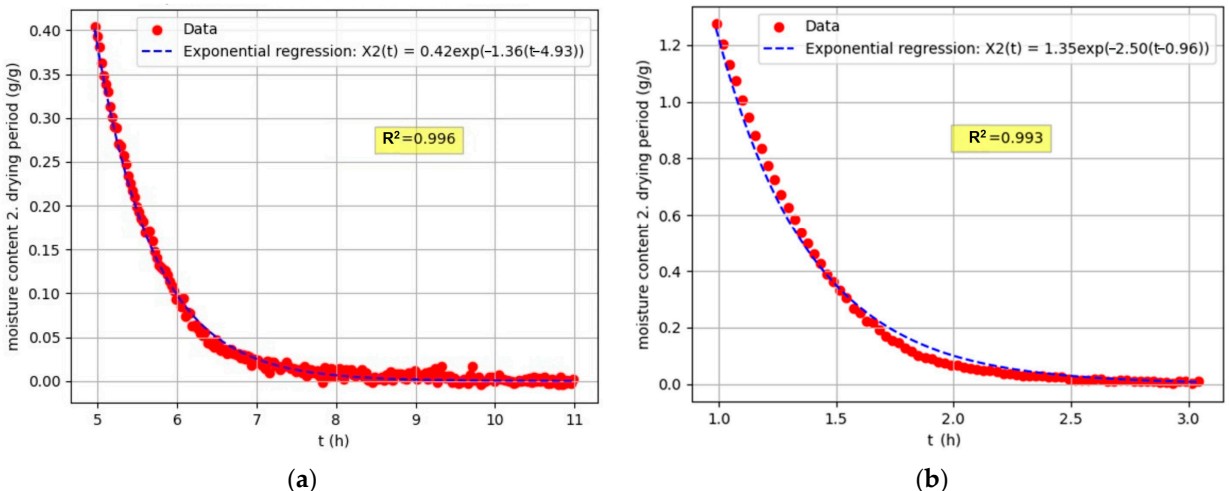

**Figure 5.** Regression functions of the second drying period: (**a**) sample 1, $T_{dry}$ = 19.4 °C, $\Psi$ = 33.6%; (**b**) sample 5, $T_{dry}$ = 52.4 °C, $\Psi$ = 8.2%.

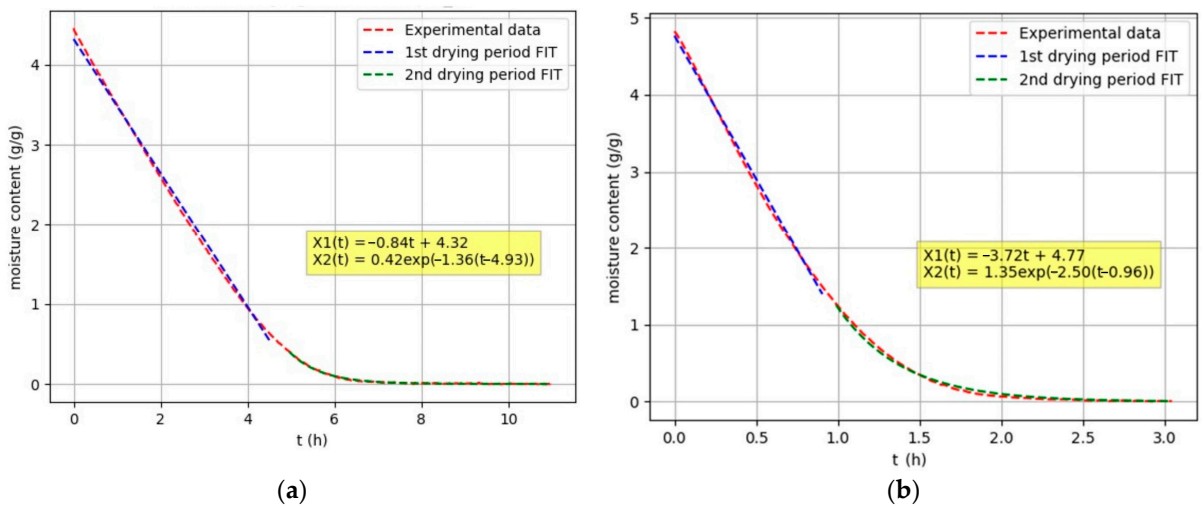

**Figure 6.** Composite function of the first and second drying periods: (**a**) sample 1, $T_{dry}$ = 19.4 °C, $\Psi$ = 33.6%; (**b**) sample 5, $T_{dry}$ = 52.4 °C, $\Psi$ = 8.2%.

Figure 6 presents a comparison between experimental data and the composite regression function of the first and second drying periods (graphs for samples 2, 3, and 4 are in the Supplementary Materials, Figures S3 and S4).

### 3.4. Stage 4 Results

The results from stage 4 are presented in Figures 7 and 8. Figure 7 displays how the first derivative changes over time during the drying process. These curves exhibit a critical point where the first derivative shifts from being constant to non-linearly increasing. The analysis of drying requires the diagrams presented in Figure 8, where the drying rate of each sample is displayed. It is noteworthy that the calculation of drying rate curves relies on mathematical regression functions. As a result, the curves in Figures 7 and 8 have a well-structured form compared to the literature on the drying of solid materials, where the presented results are solely based on experimental data and the derivation is numerically calculated. It is worth noting that a derivative curve calculated based on numerical methods will have a more irregular shape than a derivative curve calculated based on a regression function (graphs for samples 2, 3, and 4 are in the Supplementary Materials, Figures S5 and S6).

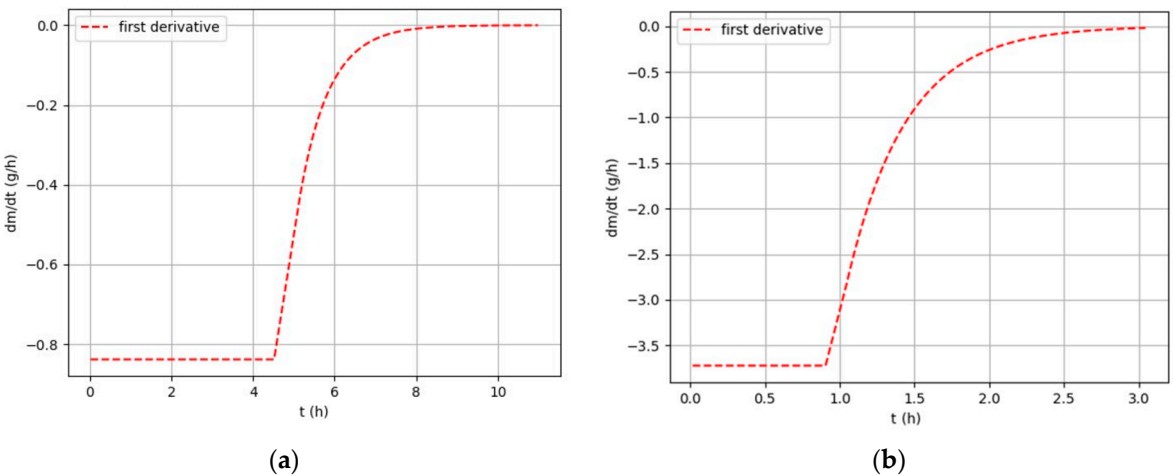

| (a) | (b) |
|:---:|:---:|

**Figure 7.** First derivative functions: (**a**) sample 1, $T_{dry}$ = 19.4 °C, $\Psi$ = 33.6%; (**b**) sample 5, $T_{dry}$ = 52.4 °C, $\Psi$ = 8.2%.

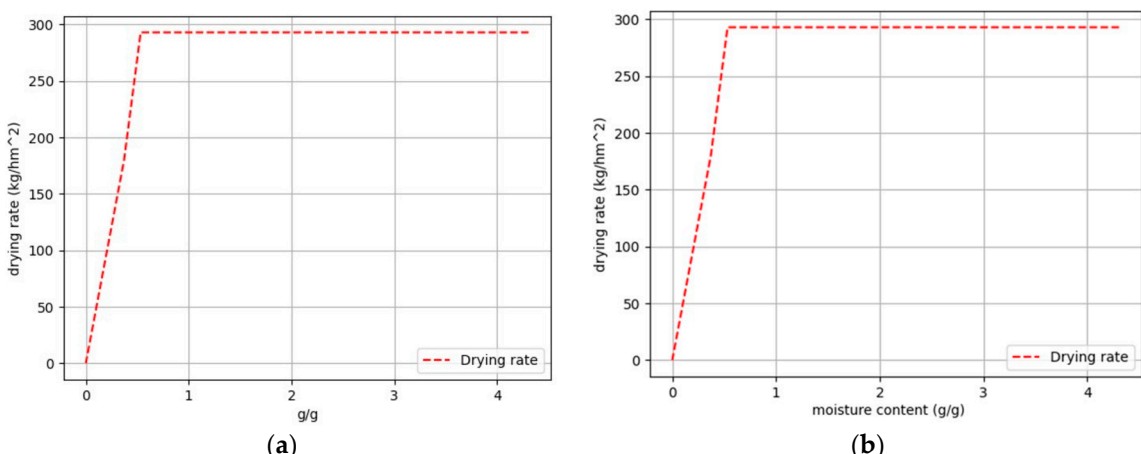

| (a) | (b) |
|:---:|:---:|

**Figure 8.** Drying curves: (**a**) sample 1, $T_{dry}$ = 19.4 °C, $\Psi$ = 33.6%; (**b**) sample 5, $T_{dry}$ = 52.4 °C, $\Psi$ = 8.2%.

### 3.5. Stage 5 Results

The normalized drying rate curves for each sample obtained from stage 5 are presented in Figure 9. These curves exhibit some degree of variation among the individual samples. Notably, the normalized drying rate curve of sample 1 displays the most pronounced deviation when compared to the others, which appear to align more closely with the expected behavior. Several factors may have contributed to the observed deviation in the drying rate curve of sample 1. First, it is possible that the equipment and methods employed to measure the equilibrium moisture content possess inherent limitations in terms of sensitivity or accuracy. Such limitations can influence the precision of the determination and introduce discrepancies in the data. In addition, variations in environmental conditions, measurement equipment, or procedural errors may have introduced greater experimental variability in the case of sample 1. These external influences and sources of variability can manifest as deviations in the drying rate curve and complicate the accurate determination of equilibrium moisture content. Furthermore, it is important to consider the impact of drying temperature and relative humidity conditions on the observed deviation. Sample 1 was subjected to the lowest drying temperature and the highest relative humidity conditions, resulting in a drying procedure that took approximately three times longer compared to, for example, sample 5. This prolonged drying duration increases the potential for error, as variations in the drying conditions over an extended period can significantly affect the equilibrium moisture content determination.

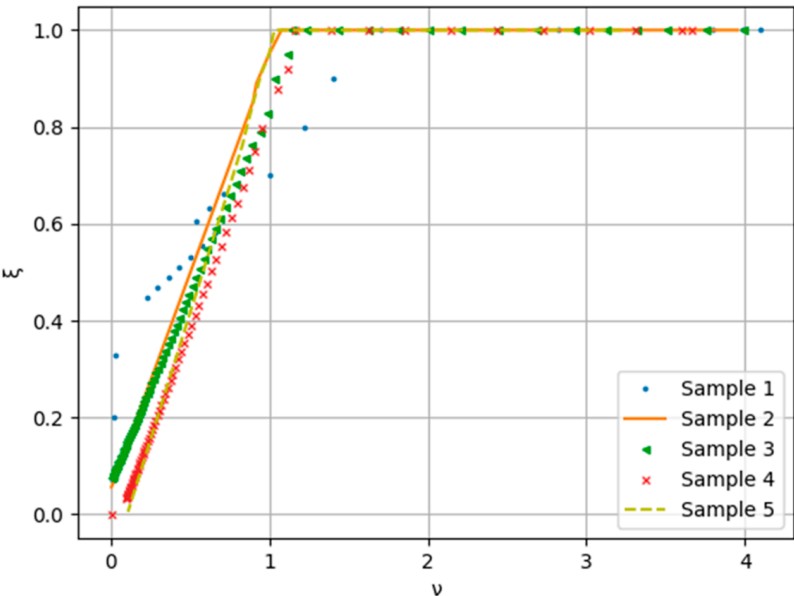

**Figure 9.** Normalized drying rate curves.

In summary, the deviation in the drying rate curve of sample 1 can be attributed to a combination of factors, including limitations in measurement equipment, procedural variations, and the significant differences in drying conditions. Recognizing the influence of these variables is essential for comprehending the observed deviations and improving the precision of future determinations.

### 3.6. Overall Results

This section aims to present and discuss the results which the software has exported to an Excel document after analyzing the data. Table 3 presents the main results of samples 1–5.

**Table 3.** Program-calculated data for sample 1–5.

| Sample | $N_a$ [gs$^{-1}$m$^{-2}$] | $q$ [W] | $h$ [Wm$^{-2}$K$^{-1}$] | $k_y$ [mol/m$^2$s] | $k'_y$ [m s$^{-1}$] | $X_{crit}$ [g g$^{-1}$] | $t_{crit}$ [h] | $D_{ab}$ [m$^2$s$^{-1}$] |
|---|---|---|---|---|---|---|---|---|
| 1 | 0.08 | 1.14 | 22.59 | 0.80 | 0.018 | 0.40 | 4.93 | $4.84 \times 10^{-10}$ |
| 2 | 0.14 | 1.96 | 31.06 | 1.07 | 0.025 | 1.07 | 2.60 | $5.04 \times 10^{-10}$ |
| 3 | 0.21 | 2.89 | 33.23 | 1.23 | 0.028 | 0.95 | 2.43 | $5.56 \times 10^{-10}$ |
| 4 | 0.29 | 3.99 | 29.88 | 1.01 | 0.024 | 1.22 | 1.35 | $7.46 \times 10^{-10}$ |
| 5 | 0.36 | 5.06 | 31.16 | 1.06 | 0.025 | 1.35 | 0.96 | $1.04 \times 10^{-9}$ |

All of the results of the individual parameters are shown in Figure 10 and are essential for understanding the drying process of MSS under forced convection on a flat plate surface. The heat transfer coefficient is the first of the important drying parameters shown in Figure 10a. The coefficient initially increases with temperature but stabilizes as the temperature increases (around 40 °C). Its average value is 29.6 W m$^{-2}$K$^{-1}$. The values were also checked using an online calculator [29]. The values calculated by the calculator refer exclusively to the heat transfer coefficient of water (liquid–vapor) and are in good correlation with the calculated values. Also, a comparison with other world literature [30], which states that the value of the heat transfer coefficient for convective drying of MSS at an air flow 0.6–2.0 m s$^{-1}$ and temperature 100–160 °C is between 21.44–40.92 W m$^{-2}$K$^{-1}$, demonstrates compatibility with the results of the software calculations. Although we have used a slightly lower drying temperature in our case, it can be assumed that the value of the heat transfer coefficient will not vary too much due to this temperature difference.

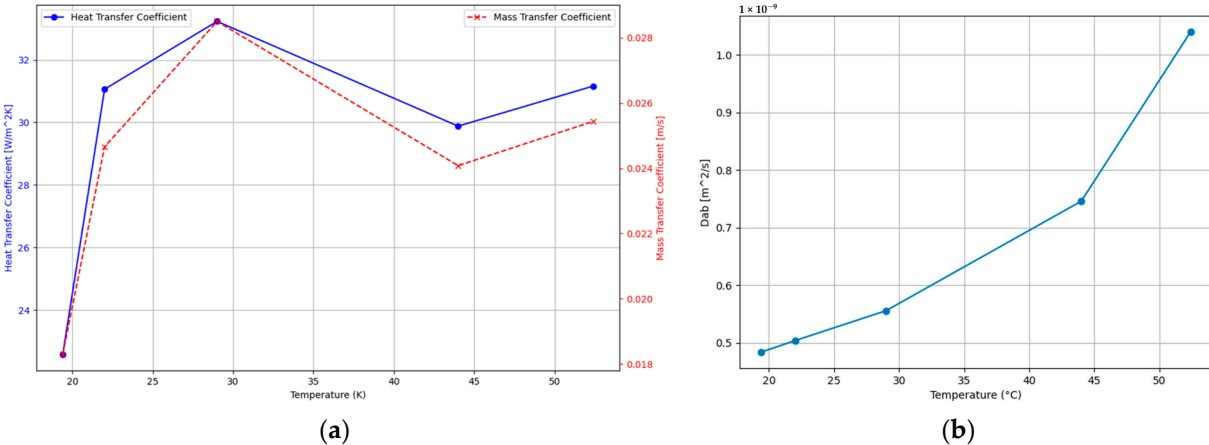

(**a**)         (**b**)

**Figure 10.** Program-calculated drying parameters as a function of temperature: (**a**) heat transfer coefficient and mass transfer coefficient; (**b**) diffusivity coefficient.

The following important drying parameter that the software calculates is the mass transfer coefficient, calculated by Equation (32). Since the heat and mass transfer coefficients are linearly dependent on each other, the shapes of the graphs in Figure 10a are almost identical. Initially, the value of the mass transfer coefficient increases with temperature and then begins to fluctuate. It is difficult to conclude from the results whether the value really increases with temperature, but we can speak of an average value of 0.024 m s$^{-1}$. This value is relatively high compared to other literature. A study [31] calculated the value of the mass transfer coefficient for convective drying at different drying air velocities and temperatures. The theoretically calculated value recorded was between 0.003–0.01 m s$^{-1}$, which is slightly lower compared to the results calculated by Python, which are between 0.018–0.028 m s$^{-1}$. The difference could be due to different drying conditions, especially relative humidity. In our case, the MSS was dried at relatively low relative humidity values (8.2–33.6%) which allowed faster drying and consequently, a higher material transfer coefficient. However,

as can be seen in article [31], the relative humidity of the drying air during the process is always above 40%.

The following drying parameters such as critical free moisture, critical drying time, mass flux of the first drying period, and heat flux of the first drying period are discussed in the Supplementary Materials.

The last drying parameter is the diffusion coefficient, which is calculated by Python from the exponential interpolation of the data in the second drying period according to the following equation:

$$D_{\text{AB}} = \frac{LV_1 k_2}{A} \tag{35}$$

where:

$L$— is the film thickness of 1.5 mm.

Note that $V_1 = AL$. The equation can be simplified as follows:

$$D_{\text{AB}} = L^2 k_2 \tag{36}$$

The calculated values of the diffusion coefficient are shown in the diagram in Figure 10b. The values of the diffusion coefficient increase with temperature, which can be confirmed by the Arrhenius equation.

### 3.7. Comparative Analysis of the Results with Existing Literature

From the graph in Figure 10b the calculated values of the diffusion coefficient range between orders of magnitude $-9$ to $-10$, which is slightly lower compared to the rest of world literature, which claims that the value of the coefficient is in the order of magnitude $-8$ to $-9$. For example, the literature gives calculated diffusion coefficient values between $5.112 \times 10^{-9}$–$1.229 \times 10^{-8}$ at an average drying temperature of 40 °C. In this study, drying experiments of direct and indirect natural convection solar drying of sewage sludge were carried out in Algeria [8]. In the following study [32], in which the value of the diffusion coefficient was determined experimentally based on MMS drying in a laboratory-scale hot air-forced convection dryer, the investigation was conducted at hot air temperatures between 100 and 160 °C and hot air velocities of 0.6, 1.4, and 2.0 m s$^{-1}$. The results of this study showed that the value of the diffusion coefficient for the second falling rate period was $1.15 \times 10^{-8}$ to $4.40 \times 10^{-8}$ m$^2$ s$^{-1}$. Some scientists have shown [33] that when drying with microwave convective dryer temperatures of 40, 55, and 70 °C and air velocities of 0.5, 1, and 1.5 m s$^{-1}$, the average value of the diffusion coefficient is $1.71 \times 10^{-9}$. This value is the closest to our calculations. The material's structure and drying circumstances may have contributed to the calculated diffusion coefficient's average value of $6.67 \times 10^{-10}$. The drying temperature in our case is in the range of 19.4–52.4 °C, whereas in the literature cited above, the range is 40–160 °C. A higher temperature therefore means a higher diffusion and diffusion coefficient (according to Arrhenius equation).

Through our comparative analysis of the results obtained using a range of programming languages and software, including Matlab [34,35], Fortran [36], COMSOL Multiphysics [37], Sigma Plot [38], TRNSYS [39], and Python, it is clear that these tools effectively calculate fundamental drying parameters such as drying rates and free moisture content. The drying data are presented in various charts and diagrams. A notable similarity among these platforms is their ability to compute the diffusivity coefficient, having values usually within the order of $10^{-10}$ to $10^{-11}$. Nevertheless, an exceptional aspect of our findings is the presentation of standardized drying rate curves for each sample. Such standardization permits a more lucid comparison and assessment of drying rates, thereby enabling an overall comprehension of the drying procedure's attributes. Additionally, our research delved deeper, enabling us to compute additional drying parameters such as mass transfer coefficient, heat transfer coefficient, and diffusion coefficient. Unlike other literature, which frequently presents a single parameter or no parameters at all, we provide a more comprehensive analysis. Moreover, our program can ascertain regressions for both the initial and secondary drying phases, as well as identify the critical point's coordinates.

The identification of the critical point serves as a crucial marker for the transition from one drying phase to another, providing valuable understanding of the drying process. Unlike previous literature, where there is little to no mention of the determination of this point, acknowledging its significance is essential. Notwithstanding, it is important to recognize certain constraints of Python in our context. Python may be less efficient for computationally intensive simulations compared to specialized languages such as Fortran or dedicated platforms like COMSOL Multiphysics. Furthermore, Python may exhibit lower performance with significantly large datasets than software optimized for specific engineering applications, like TRNSYS.

## 4. Conclusions

This study shows that the Python programming language is effective in analyzing and modeling chemical processes, particularly the drying process. A successful analysis of drying data was accomplished using Python, and a program was created that could be beneficial in developing chemical processes on a laboratory scale. The significant drying parameters, such as the mass transfer coefficient, heat transfer coefficient, diffusion coefficient, drying rate, critical humidity, and critical drying time, were determined by the program. Furthermore, this research proves that Python is a suitable tool for modeling the drying process. The software analyzed the initial drying phase by utilizing a linear equation (Equation (2)). It also automatically detected the point where the drying rate decreases, signifying the start of the second drying period. An exponential function (Equation (3)), serving as a mathematical model, demonstrated exceptional conformity with the experimental data ($R^2 > 0.99$) during this phase. Thereafter, a master curve for five different MSS samples was developed through analysis. This curve is instrumental in the planning of a drying process at different temperatures. This research showcases how Python can be a powerful tool for the analysis and modeling of chemical processes. The program devised in this research can be employed to draft and refine chemical procedures on an extensive scale. Moreover, the master curve can be utilized to blueprint the drying procedure under diverse temperatures. Furthermore, future investigations could enhance the precision of the master curve through the creation of a more advanced model or the inclusion of additional data from diverse MSS samples.

**Supplementary Materials:** The following supporting information can be downloaded at: https://www.mdpi.com/article/10.3390/pr11123263/s1, Figure S1: Mass of MSS over time: (a) Sample 2, $T_{dry}$ = 22.0 °C, $\Psi$ = 22.4% (b) Sample 3, $T_{dry}$ = 29.0 °C, $\Psi$ = 14.8% (c) Sample 4, $T_{dry}$ = 44.0 °C, $\Psi$ = 10.0%; Figure S2: Regression functions of the first drying period: (a) Sample 2, $T_{dry}$ = 22.0 °C, $\Psi$ = 22.4% (b) Sample 3, $T_{dry}$ = 29.0 °C, $\Psi$ = 14.8% (c) Sample 4, $T_{dry}$ = 44.0 °C, $\Psi$ = 10.0%; Figure S3: Regression functions of the second drying period: (a) Sample 2, $T_{dry}$ = 22.0 °C, $\Psi$ = 22.4% (b) Sample 3, $T_{dry}$ = 29.0 °C, $\Psi$ = 14.8% (c) Sample 4, $T_{dry}$ = 44.0 °C, $\Psi$ = 10.0%; Figure S4: Composite function of the first and second drying periods: (a) Sample 2, $T_{dry}$ = 22.0 °C, $\Psi$ = 22.4% (b) Sample 3, $T_{dry}$ = 29.0 °C, $\Psi$ = 14.8% (c) Sample 4, $T_{dry}$ = 44.0 °C, $\Psi$ = 10.0%; Figure S5: First derivative functions: (a) Sample 2, $T_{dry}$ = 22.0 °C, $\Psi$ = 22.4% (b) Sample 3, $T_{dry}$ = 29.0 °C, $\Psi$ = 14.8% (c) Sample 4, $T_{dry}$ = 44.0 °C, $\Psi$ = 10.0%; Figure S6: Drying curves: (a) Sample 2, $T_{dry}$ = 22.0 °C, $\Psi$ = 22.4% (b) Sample 3, $T_{dry}$ = 29.0 °C, $\Psi$ = 14.8% (c) Sample 4, $T_{dry}$ = 44.0 °C, $\Psi$ = 10.0%; Figure S7: Programe-calculated drying parameters as a function of temperature: (a); critical moisture content (b); critical time (c); mass flux of the 1. drying period (d); heat flux of the 1. drying period; Information S1: Pyhton Program Code.

**Author Contributions:** Conceptualization, E.M. and D.K.; methodology: E.M. and D.K.; software writing, E.M.; validation, P.O. and K.R.; writing—review and editing: D.K., P.O. and K.R. All authors have read and agreed to the published version of the manuscript.

**Funding:** The authors thank the Slovenian Research and Innovation Agency—ARIS, Project L7-3185 and Program P2-0414, for the financial support.

**Data Availability Statement:** All available instructions on the use of experimental data are now explained in the supplementary material.

**Conflicts of Interest:** The authors declare no conflict of interest.

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
