# Peer review of "Analyzing and Modeling the Municipal Sewage Sludge Drying Process Using Python"

_processes, doi:10.3390/pr11123263_

Round 1

Reviewer 1 Report

Comments and Suggestions for Authors

Authors used Python programming language in modelling sewege sludge drying samples. The topic is interesting and I believe authors presented the topic adequately.

I suggest authors the following feedback for improving the manuscript:

1- Avoid using numbers or formulas in the abstract. Abstract should be organized in a way to address the rationale, methods, findings and implications.

2- Introduction should  include brief introduction to the topic and last paragraph should be added as to describe the outline of the rest of the paper. Move stages under Materials and Methods section.

3- Could you check your equations there is an additional ',' sign by the end of most of the equations. Please remove that. 

4- Some figures are not mentioned in the text. Check all figures and tables and link them to the text. 

5-Add relevant references to the discussion section. This section should include discussing results with the existing literature.

6- could you compare and contrast the results of your study to another study or studies that have used another programming language for modeling. 

7- Could you justify from the literature that this is the only known study that have used Python language to model drying process?

8- References section  should be improved by the thorough scanning of the extant literature.

Comments on the Quality of English Language

Moderate English proof reading is required.

Reviewer 2 Report

Comments and Suggestions for Authors

The paper proposes to demonstrate the effectiveness of Python in analyzing and modeling chemical processes, using municipal sewage sludge (MSS) drying as an example. Overall, the use of typical libraries and analysis in Python may not be considered a novelty for paper publication, as it is a common practice in many research fields. If the paper addresses a novel research question, proposes a new methodology, or obtains novel results, it can be considered a novelty regardless of the tools used, which is not clearly described in this study. Therefore, there are some concerns that must be resolved before it can be considered suitable for publication. The following points outline the issues that need to be addressed:

·         The study only focuses on one specific case study of MSS drying, and it is unclear how generalizable the results are to other chemical engineering processes.

·         The study does not discuss the limitations or challenges of using Python for chemical engineering analyses and modeling.

·         The keywords selected are not suitable and cannot represent all papers in the subjects.

·         The introduction could be more concise and better structured to provide a clearer understanding of the objectives of the paper.  It is crucial to clearly what were previous studies lacking that was address in the present work. At introduction part, explain the potential significance or impact of your research. Discuss how your findings may advance knowledge, fill gaps, solve problems, or have practical implications. This emphasizes the relevance and value of your study.

·         The writing and format of the manuscript need major improvements. When writing a scientific article, the detailed explanation of how the program works and the theoretical background on which all the calculations are based should be located in the Materials and Methods section and not in the Introduction.

·         The paper does not compare the performance of Python to other programming languages or software tools commonly used in chemical engineering.

·         It is strongly recommended that the author update their references to include more recent publications, which would improve the paper's credibility and demonstrate that the author has thoroughly researched the subject matter.

Reviewer 3 Report

Comments and Suggestions for Authors

The manuscript entitled “Using Python to Analyze and Model Chemical Engineering Processes: A Case Study of MSS Drying” is interesting. However, the article has many scientific flaws, some examples of which are mentioned below. The writing standard of this manuscript should be improved. I recommend the authors rewrite the manuscript and resubmit.

The introduction section is too long, but in my opinion it is due to the fact that most of the text should be in next part of manuscript (actually, from line 76 this text refers to research methodology). I suggest The Materials and Methods section should be rewritten, and divided into different subsections, for example, preparation of samples for drying, drying, mathematical modelling, the Python software,… etc, or just use titles what are now in the introduction section.

In the introduction, the literature review is quite poor, it could contain more references regarding the drying process, as well as the mathematical modeling of this process.

The purpose and scope of the work are not clearly formulated in the manuscript.

In line 84 it is written that the experiment was carried out in the temperature range of 20 to 50°C but the results show temperatures of 19.4 and 52.4.

The procedure was conducted at five different drying temperatures ranging from 20 to 50 °C, but it was not stated how many repetitions at each temperature (if only one, why, if more, how many times). It is also worth naming these 5 temperatures specifically.

It is not clear what data is sorted (line 90, 110) and from which the average is calculated (eq 1).

Why a linear model was used to first period of drying (line 137)? It is true, that in the first drying period, the evaporation rate is mainly determined by external conditions of heat and mass transfer but the process speed does not have to be constant. According to the convective drying theory, the first period model is the result of solving the heat balance equation and, after taking into account the initial conditions and the drying rate coefficient k0 and drying shrinkage, it can also be described by the third-degree equation.

There are many second drying period models based on an exponential function, why did the authors choose this one (eq. 3)?

How the program determines the critical drying point (line 154)?

In eq (4) m0 means mass of the sample after drying but the m0 usually denotes the initial mass, and in line 171 m0 means the initial moisture content, what is not true.

To calculate the moisture content in equation (4), the mass of the dry substance should be used, not mass after drying.

Did Authors determine (and how) the mass of dry matter? because it is not written about it. This is a crucial parameter for determining the material's moisture content.

Is it necessary in this manuscript, to derive a function describing the free moisture content of a material during the second drying period?

Whether the authors derived the function of the second drying period themselves or used formulas available in the literature?

In line 249 Tdry is in [K] and then in table 1 [°C]

Does in table 1 Tdry mean the temperature of the drying air? Wouldn't it be better to call it just drying temperature? And why such particular drying temperatures?

What is Rv and Rw in eq. (33)?

In eq. 34 there is X, and previously there was always Xt, is it the same variable?

Line 317-322 As the results in Figure 5 showed, weighing variations caused by air blowing from the dryer still occur, so the method of eliminating deviations by blind measurement seems to be incorrect.

I think that the initial mass of the sample is too small, which may cause too large a measurement error

In Figure 3a is m and t, and in Fig 3b is mass and time

Line 400 I think should be Fig.7 and in line 403 Fig. 8

Line 417-418 Why the equilibrium moisture content value has not been precisely determined for sample 1

In Figure 10a the temperature is in K and in Figure 10b in °C

The second equation 35 should be number 36

Line 465 Diffusivity coefficient is in fig 10b

Figure 11 is difficult to read and incomprehensible

References: Most of the references are too old. Authors should use up-to-date references from the previous five years.

I consider the article to lack the necessary standards, so this article is not recommended for publication in a journal.

Round 2

Reviewer 1 Report

Comments and Suggestions for Authors

I'm satisfied with the current version of the manuscript.  Authors put substantial effort to improve the content.

Author Response

Response: Thank you for taking the time to review our manuscript. We genuinely appreciate your positive feedback and are delighted to hear that you are satisfied with the current version. We are dedicated to improving the quality of our work, and your comments have been instrumental in achieving that goal. Your support is greatly valued, and we look forward to any further insights you may have.

Reviewer 2 Report

Comments and Suggestions for Authors I have carefully reviewed the revised paper and responses, but unfortunately, I find some of them unsatisfactory. Specifically, the response discusses how Python can be used to analyse data from digitalized measuring equipment, but it does not address the main concept and title of the paper, which is to determine the generalizability of the results of this study to other chemical engineering processesTherefore, despite the interesting topic of the paper, it is not acceptable for publication in this format, even with some revisions.

Reviewer 3 Report

Comments and Suggestions for Authors

I would like to ask for additional explanations:

 - Line 164: if the results of mass readings were corrected for absolute error by subtracting from the data for further analysis, so on the diagrams (Fig 5), moisture content values should no drop below 0.

- Do the data referred to in lines 191, 221 and presented in Fig. 1 concern all 3 repetitions for 5 dried samples, or a selected sample at one drying temperature?

Samples dried at different temperatures have different parameters, so they should not be on one chart, and if these are data for an example sample, it should be written for which one.

 - Line 263/ eq 4: m0 usually refers to the initial mass and calling it “mass of dry matter/substance” may be misleading for the reader. I would suggest though changing this designation.

Additionally, the term "dry mass of dry matter" also appears in formula 22 (line 343) and has the symbol Ls

The same parameter should be marked with the same letter throughout the manuscript

 - Line 477: “The deviation in the curve of sample 1 can be attributed to a combination of factors. Firstly, the equipment and methods used to measure the equilibrium moisture content may have limitations in sensitivity or accuracy, affecting the accuracy of the determination.”

Was different equipment and methods used to measure the equilibrium moisture value of 1 sample than to determine the moisture content of other samples? Because there was no such problem for other samples dried at other temperatures.

Round 3

Reviewer 2 Report

Comments and Suggestions for Authors

Still, I think that title of the paper does not accurately reflect its content. Therefore, I propose revising the title to "Analyzing and modeling  the municipal sewage sludge drying using Python". This revised title is more specific and informative, as it accurately reflects the focus of the study and can help readers understand the significance of the research.
The abstract and other parts of the paper should be revised accordingly.

Author Response

Title, abstract, and other parts of the paper have been revised accordingly to reflect the new title, "Analyzing and modeling the municipal sewage sludge drying using Python."